# The Obesity Amelioration Effect in High-Fat-Diet Fed Mice of a Homogeneous Polysaccharide from *Codonopsis pilosula*

**DOI:** 10.3390/molecules27165348

**Published:** 2022-08-22

**Authors:** Qi Su, Jiangyan Huo, Yibin Wang, Yang Zhou, Dan Luo, Jinjun Hou, Zijia Zhang, Huali Long, Xianchun Zhong, Cen Xie, Min Lei, Yameng Liu, Wanying Wu

**Affiliations:** 1School of Chinese Materia Medica, Nanjing University of Chinese Medicine, Nanjing 210023, China; 2Shanghai Research Center for Modernization of Traditional Chinese Medicine, Shanghai Institute of Materia Medica, Chinese Academy of Sciences, Shanghai 201203, China; 3National Engineering Laboratory for TCM Standardization Technology, Shanghai Institute of Materia Medica, Chinese Academy of Sciences, Beijing 100049, China

**Keywords:** *Codonopsis pilosula*, polysaccharide, structural characterization, obesity

## Abstract

A homogeneous polysaccharide coded as CPP−1 was extracted and purified from the root of *Codonopsis pilosula* (Franch.) Nannf. by water extraction, ethanol precipitation, and column chromatography. Its structure was analyzed by HPGPC-ELSD, HPLC, GC-MS, FT-IR, and NMR techniques. The results indicated that CPP−1 was composed of mannose (Man), glucose (Glc), galactose (Gal), and arabinose (Ara) at a molar ratio of 5.86 : 51.69 : 34.34 : 8.08. The methylation analysis revealed that the main glycosidic linkage types of CPP−1 were (1→)-linked-Glc residue, (1→3)-linked-Glc residues, (1→4)-linked-Gal residue, (1→2,3,4)-linked-Glc residue, (1→)-linked-Man residue, (1→3,4)-linked-Glc residue, and (1→)-linked-Ara residue. In vivo efficacy trial illustrated that CPP−1 supplements could alleviate HFD-induced mice obesity significantly, as well as improve obesity-induced disorders of glucose metabolism, alleviate insulin resistance, and improve the effects of lipid metabolism. The findings indicate that this polysaccharide has the potential for the treatment of obesity.

## 1. Introduction

Obesity is an epidemic and a major public health problem prevalent worldwide [1], which can cause or exacerbate many health problems, such as metabolic syndrome, cardiovascular disease, sleep and breathing disorders [2,3,4]. It is therefore important to accelerate the prevention and treatment of obesity. At present, dietary intervention and pharmacotherapy are common choices for long-term weight reduction, but they are usually largely compromised mainly due to poor compliance and side effects [5,6]. On the contrary, it is worth noting that polysaccharides effectively prevent or treat obesity with few side effects [7,8,9].

Polysaccharides, as biological macromolecules existing widely in nature, have a variety of physiological functions and biological activities [10]. It has been demonstrated that polysaccharides from Traditional Chinese medicine, such as *Lycium barbarum* [11], *Morus alba* L. [12], *Ganoderma amboinense* [13], and *Dioscorea opposite* [14], can prevent or suppress obesity. Considering the well-recognized anti-obesity potentials of polysaccharides [8], *Codonopsis pilosula* polysaccharides (CPS) may serve as another class of natural anti-obesity agents [15]. Studies have shown that CPS contribute to the improvement of metabolic syndrome [16,17,18]. For example, CPS significantly lowered blood glucose [16] and significantly improved lipid metabolism disorders in diabetic rats [19]. Obesity, like diabetes, is a metabolic syndrome [20]. As far as we know, studies on the biological activity of purified CPS, especially in anti-obesity has been rarely explored.

Radix Codonopsis is the dried root of *Codonopsis pilosula* (Franch.) Nannf., *C. pilosula* Nannf. var. *modesta* (Nannf.) L. T. Shen or *C. tangshen* Oliv. Radix Codonopsis has a wide range of branches in China and a long history of medicinal use. Of the above three kinds of Radix Codonopsis, *C**. pilosula* is the most widely used in clinical practice [21]. The traditional effects of *C**. pilosula* in China are to strengthen the spleen, benefit the lungs, nourish the blood and promote the production of body fluid [22]. Recent research in natural medicinal chemistry has revealed that the main bioactive substances in the root of *C**. pilosula* contain polysaccharides [23], alkaloids [24], flavonoids [25], lignans [26], polyacetylene [27,28], terpenoid [29,30], and lactones [31]. Among those, polysaccharides are one of the major active components, and related research has attracted great interest [23]. In the past few years, CPS have shown prospects in treating of various diseases or as a functional ingredient in food products. It has various pharmacological activities, such as antiviral [32], antitumor [33], antioxidant [34], hypoglycemic [16], immunomodulatory [35], prebiotic [36], and neuroprotective effects [37]. Therefore, it is meaningful to study the isolation, purification, and biological activities of polysaccharides from *C**. pilosula.*

In the current study, a homogeneous polysaccharide named CPP−1 was obtained from the *C**. pilosula*. The structural characteristics of CPP−1 were analyzed by high-performance gel permeation chromatography (HPGPC), Fourier transform infrared spectroscopy (FT-IR), gas chromatography and mass spectrometry (GC-MS), and nuclear magnetic resonance (NMR). The present study examined the anti-obesity potentials of polysaccharides isolated from *C**. pilosula* in high-fat-diet (HFD)-induced obese mice.

## 2. Results and Discussion

### 2.1. Extraction, Isolation, and Purification of CPP−1

The crude polysaccharide CPP80 (96.3 g, yield: 1.61%) was extracted with hot water, precipitated with 80% ethanol, and deproteinized by the Sevage method. CPP80 was further purified by DEAE-52 cellulose, which was eluted with distilled water and different concentrations of stepwise NaCl solution at a flow rate of 15.0 mL/min. Moreover, the fraction eluted by distilled water was named CPP80−A.

The elution curve is displayed in Figure 1A. By eluting through a Sephacryl S-200 HR gel column (2.5 cm × 100 cm), a homogenous polysaccharide CPP−1 was finally obtained. As shown in Figure 1B, CPP−1 showed a symmetrical peak, indicating it was a homogeneous polysaccharide and the molecular weight was 2.8 kDa according to the standard curve (lgM_p_ = −0.3457 t_R_ + 11.084).

### 2.2. Monosaccharide Composition Analysis of CPP−1

After the CPP−1 was completely acidic hydrolyzed, its monosaccharide composition was determined by HPLC with a DAD detector, and the monosaccharide standards were used as control. The monosaccharide and acid analysis (Figure 1C) revealed that CPP−1 was mainly composed of mannose (Man), glucose (Glc), galactose (Gal), and arabinose (Ara) at a molar ratio of 5.86 : 51.69 : 34.34 : 8.08. The monosaccharide composition experiments showed that the Gal and Glc were the main components of CPP−1.

### 2.3. Methylation Analysis

Methylation and GC-MS analysis were conducted to further study the glycosides bonding information of CPP−1. The hydroxyl groups of the polysaccharide were methylated, reduced, hydrolyzed, and acetylated to partially methylated alditol acetates (PMAAs). The types of glycosidic residues of the sample were determined by GC-MS. The GC-MS result of CPP−1 is shown in Table 1. Based on the characteristic ion fragments in GC-MS and CCRC database, it showed the presence of seven components (Appendix A), which are named 2,3,4,6-Me_4_-Glc*p*, 2,4,6-Me_3_-Glc*p*, 2,3,6-Me_3_-Gal*p*, 6-Me-Glc*p*, 2,3,4,6- Me_4_-Man*p*, 2,6-Me_4_–Glc*p*, and 2,3,5-Me_3_-Ara*f*. The glycosidic linkage types of CPP−1 were confirmed as T-Glc*p*, 1,3-Glc*p*, 1,4-Gal*p*, 1,2,3,4-Glc*p,* 1-Man*p*, 1,3,4-Glc*p*, and T-Ara*f* in a ratio of 20.53 : 4.80 : 34.84 : 10.36 : 5.17 : 15.25 : 9.05. The methylation results showed that Glc and Gal residues were the main components of the CPP−1 main chain. The results were basically consistent with monosaccharide composition analysis.

### 2.4. Fourier Transform Infrared Spectroscopy (FT-IR) Analysis

IR spectroscopy is a technique for identifying the characteristic organic groups in polysaccharides. The intense absorption band near 3600–3200 cm^−1^ indicated intermolecular and internal hydrogen bonds, indicating the characteristic spectrum of the main substituent on the sugar chain [38]. The FT-IR spectrum of CPP−1 displayed a typical spectrum of the polysaccharide (Figure 1D). The strong wide absorption band at 3241 cm^−1^ represented the stretching vibration of O-H in the constituent sugar residues [39], and the bands at approximately 2921 cm^−1^ and 1457 cm^−1^ represented C-H stretching and bending vibrations in the sugar ring [40]. The representative absorptions around 1019 cm^−1^ demonstrated the presence of the pyranose ring.

### 2.5. Nuclear Magnetic Resonance (NMR) Spectroscopy

The structural features of CPP−1 were further investigated by NMR analysis. Typically, the anomalous signal in ^1^H NMR spectrum is concentrated in the range of 4.4–5.8 ppm, where the 5.0–5.8 ppm region represents the type of α-configuration and 4.4–5.0 ppm region represents the β-configuration. Other hydrogen signals were concentrated in the 3.0–4.5 ppm. The ^1^H and ^13^C NMR spectra of CPP−1 are shown in Figure 2A,B. In the ^1^H NMR spectrum, CPP−1 had seven anomeric proton signals at 4.90, 4.90, 5.07, 5.17, 5.24, 5.35, and 5.35 ppm. Seven anomeric carbon signals were detected in the ^13^C NMR spectrum, which were at 97.79, 91.56, 107.44, 92.43, 100.25, 92.48, and 90.53 ppm. The results of ^1^H and ^13^C NMR indicated CPP−1 possessed α-configuration and β-configuration. The chemical shifts of these anomeric regions in the HSQC spectrum were 4.90/97.79 ppm, 4.90/91.56 ppm, 5.07/107.44 ppm, 5.17/92.43 ppm, 5.24/100.25 ppm, 5.35/92.48 ppm, 5.35/90.53 ppm.

### 2.6. Effect of CPP−1 Supplementation on Body Weight Gain in HFD-Fed Mice

High-fat diet are considered a contributing factor to obesity [41]. To investigate whether CPP−1 may contribute to ameliorating phenotypes of obesity, we established the obesity model using HFD for 8 weeks before CPP−1 treatment. After 8 weeks’ feeding, HFD mice were randomized and assigned to HFD group and CPP−1 supplementation group (HFD + CPP−1). The HFD + CPP−1 group were then subjected to daily intragastric treatment with CPP−1 for another 8 weeks. As shown in Figure 3A, HFD-mice group showed significantly increased body weight compared to the control group, indicating the successful establishment of the obesity model. Meanwhile, CPP−1 treatment reduced body weight gain in HFD-fed mice without any significant difference in food intake (Figure 3A,B). In Figure 3C, CPP−1 treatment reduced epididymal white adipose tissue (eWAT) weight but had no significant effect on inguinal white adipose tissue (iWAT) and brown adipose tissue (BAT). These results suggested that the oral administration of CPP−1 prevents obesity.

### 2.7. CPP−1 Supplementation Improves Blood Glucose Metabolism

Extensive evidence has shown that obesity is associated with abnormal glucose tolerance [42]. CPP−1 treatment significantly lowered random blood glucose in HFD-fed mice (Figure 3D), indicating the improved glucose metabolism effect of CPP−1 supplementation. To further investigate the effect of CPP−1 on obesity-related glucose metabolism dysfunction, the glucose tolerance test (GTT) was performed in three groups of mice. As shown in Figure 3E, after orally gavage glucose, the HFD group displayed a remarkably higher level of blood glucose than the control group, while the impaired glucose tolerance was reversed by CPP−1 supplementation. The fasting blood glucose (FBG) level of control group is 4.06 mmol/L (mean value) while the FBG level of HFD elevated to 4.94 mmol/L (mean value), 21.6% increasingly compared to the control group. Although no statistical significance was observed, CPP−1 treatment group showed 10% lower FBG than the HFD group (Figure 3G). The levels of fasting insulin were significantly different among the three groups and the HFD + CPP−1 group exhibited a significantly lower fasting insulin level (Figure 3H), indicating ameliorating effects on insulin resistance in HFD + CPP−1 group. However, in the HFD group, the mean HOMA-IR index is as high as 74.5, indicating severe insulin resistance. Compared to the HFD group, the HFD + CPP−1 group displayed lower HOMA-IR index (mean value = 28.4), confirming the insulin improvement effect of CPP−1 treatment (Figure 3I). These results revealed that the CPP−1 could improve the obesity-induced impairment of glucose metabolism and alleviate insulin resistance.

### 2.8. CPP−1 Supplementation Improves Lipid Metabolism

After 16 weeks of HFD treatment, the triglyceride (TG), total cholesterol (T-CHO), low-density lipoprotein cholesterol (LDL-C), high-density lipoprotein cholesterol (HDL-C), alanine aminotransferase (ALT), aspartate aminotransferase (AST) levels in the serum and the triglyceride, total cholesterol levels in the liver were significantly stimulated by HFD feeding (Figure 4A–H). CPP−1 treatment reversed the HFD-induced elevation of serum total cholesterol (Figure 4B), ALT (Figure 4E), AST (Figure 4F), and hepatic triglyceride (Figure 4G). These results suggested the lipid metabolism improvement effect of CPP−1.

### 2.9. CPP−1 Supplementation Alleviates Lipid Accumulation in the Liver and eWAT

As shown in Figure 4I, hematoxylin and eosin (H&E) staining of liver sections showed that hepatic steatosis and lipid droplet accumulation in obese mice were effectively decreased after CPP−1 treatment. Meanwhile, CPP−1 treatment decreased the adipocyte size in the eWAT. These results confirmed the positive effect of CPP−1 in obese mice.

## 3. Materials and Methods

### 3.1. Materials

The dried roots of *Codonopsis pilosula* were collected from Livzon Pharmaceutical Grouping Limin Pharmaceutical Factory. (P.R. China). Standard monosaccharides including Ara, Gal, Glc, Man, rhamnose (Rha), xylose (Xyl), glucuronic acid (GlcA), and galacturonic acid (GlaA) were purchased from Sigma (Louis, MO, USA). DEAE-52 cellulose was purchased from Whatman (Kent, UK). 1-Pheny-3-methyl-5-pyrazolone (PMP), dextrans (2,000,000, 300,600, 135,350, 64,650, 36,800, 13,050, 9750, 5250, 2700, and 180 Da) were purchased from the national institutes for food and drug control (Beijing, China). All other chemicals were of analytical grade.

### 3.2. Extraction, Purification and Fractionation of Polysaccharides

Dried *C**. pilosula* roots (6 kg) were cut into 1–2 cm long segments. After extraction by refluxing 80% ethanol to remove low molecular weight and lipophilic compounds, the residue was dried for further studies. The dried material was extracted with 8 volumes of distilled water at 100 °C for 3 h for three times. The whole extract supernatant was combined, filtered, and concentrated. The concentrated aqueous extract was precipitated with 95% ethanol to a final concentration of 80% and deproteinized using the Sevage method [43] to obtain crude polysaccharide (CPP80). CPP80 was applied to a DEAE cellulose-52 ion-exchange chromatography column eluted sequentially with distilled water and a linear gradient of NaCl solutions (0.1, 0.2, 0.4, 0.8 M) to obtain CPP80-A, CPP80-B, CPP80-C, and CPP80-D. CPP80-A (16 g) was dissolved in distilled water and injected into a column (2.5 × 100 cm) of Sephacryl S-200 HR. The column was eluted with distilled water at 0.2 mL/min (15 min/tube). All fractions were monitored spectrophotometrically using the phenol-sulfuric acid method [44]. The main fractions were collected, concentrated, dialyzed, and lyophilized to produce CPP−1 (4 g).

### 3.3. Homogeneity Evaluation and Molecular Weight Determination

Homogeneity and molecular weight of CPP−1 were determined by HPGPC using an Agilent 1260 (Santa Clara, CA, USA) equipped with an ELSD. The contents of carbohydrate were determined by the phenol-sulfuric acid assay method [45].

### 3.4. Monosaccharide Composition

The monosaccharide composition of CPP−1 was determined by PMP pre-column derivatization with HPLC [46]. CPP−1 (2.0 mg) was hydrolyzed by 1 mL of 4 M trifluoroacetic acid (TFA) at 110 °C for 5 h. A rotatory evaporator was used to remove excess acid using methanol, and the hydrolyzed product was dissolved in 200 µL ammonium solution and mixed with 200 µL of 0.5 M PMP-methanol solution, and the reaction was conducted at 70 °C for 1.5 h. After the elimination of ammonia by nitrogen, the reaction product was dissolved in 200 µL water and extracted with 200 µL chloroform three times. Finally, the supernatant was tested on Agilent 1260 HPLC equipped with an Agilent C18 column (4.6 × 200 mm, 5 µm). The sample was eluted with 20 mM ammonium acetate in water (82.5%) and acetonitrile (17.5%) at 25 °C with 1 mL/min flow rate.

### 3.5. Fourier-Transform Infrared (FT-IR) Spectroscopy

The IR spectrum of CPP−1 was measured using Thermo Nicolet iS5 FTIR spectrometer (Waltham, MA, USA). CPP−1 (2 mg) dissolved in chloroform was added on the KBr window and dried under infrared light, which was measured in the middle infrared range of 4000–400 cm^−1^.

### 3.6. Methylation Analysis

The methylation analysis of CPP−1 was referred to as the Hakomori method with some modification [47,48]. The sodium methyl sulfinyl methyl (SMSM) was prepared by slowly adding NaH to anhydrous DMSO and heating for 4–6 h. CPP−1 (10 mg) was per-O-methylated by repeated reaction with SMSM and methyl iodide until the methylation was completed. Complete methylation of CPP−1 was confirmed by no obvious peak around 3500 cm^−1^ in the FT-IR spectrum. After that, the residue was hydrolyzed by 2 M TFA at 120 °C for 6 h. Then the sample was reduced with NaBD_4_ (25 mg) for 30 min at 40 °C. The sample was acetylated with acetic anhydride (2 mL) and pyridine (2 mL). Finally, the partially methylated alditol acetates (PMAAs) were analyzed by GC-MS.

### 3.7. Nuclear Magnetic Resonance (NMR) Spectroscopy Analysis

CPP−1 (60 mg) was dissolved in deuterium oxide (D_2_O, 500 µL), and exchanged hydrogen with deuterium three times by repeated freeze-drying. After lyophilization, the sample was dissolved in D_2_O (500 µL), centrifuged and transferred into an NMR tube. The NMR spectra (^1^H, ^13^C, HSQC, and HMBC) were obtained by a 500 MHz Bruker spectrometer (Karlsruhe, Germany).

### 3.8. Animal Studies

C57BL/6 male mice (eight-week-old) used in this experiment were obtained from HuaFukang BioScience Company (Beijing, China) and housed in SPF-grade according to requirements of the Institutional Ethics Committee of Shanghai Institute of Materia Medica (2021-04-XC-38). Mice were kept in a temperature-controlled room at 23 ± 2 °C with 30–70% relative humidity. The light/dark cycle was 12/12 h, and the mice had free access to water and food.

Before the study started, mice were acclimatized on standard food for 1 week and then randomly allocated to three groups (n = 10 for HFD group and n = 5 for Con group). Mice in the normal control group (Con group) were fed with a low-fat diet (Research Diets, D12450J, 10% kcal fat), and HFD group were fed with a high-fat diet (Research Diets, D12492, 60% kcal fat). After 8 weeks’ feeding, HFD mice were randomized and assigned to HFD group and CPP−1 treatment group (HFD + CPP−1). The HFD + CPP−1 group were then subjected to daily intragastric treatment with 200 mg/kg CPP−1 for another 8 weeks. Throughout the experiment, each mouse’s body weights were recorded weekly and each cage’s food intakes were recorded daily.

### 3.9. Random Glucose Test

Random glucose was measured at any time with blood samples taken from tail veins using blood sugar test paper glucometer (Sannuo Biosensors, Changsha, China).

### 3.10. Glucose Tolerance Test

The mice were fasted for 16 h and oral gavage of D-glucose (2 g/kg body weight). Blood glucose levels at 0, 15, 30, 60, 90, and 120 min post glucose administration were measured with blood samples taken from tail veins using blood sugar test paper and glucometer (Sannuo Biosensors, Changsha, China).

### 3.11. Fasting Blood and Fasting Insulin Assays

Fasting blood glucose was measured after 16 h fasting with blood samples taken from tail veins using blood sugar test paper and glucometer (Sannuo Biosensors, Changsha, China). Fasting serum insulin were tested using mouse insulin ELISA kit (Crystal Chem, IL, USA). Homeostatic model assessment-insulin resistance (HOMA-IR) index was calculated using formula:

HOMA-IR = fasting blood glucose (mmol/L) × fasting insulin (mIU/L)/22.5.

### 3.12. Biochemical Analyses

Triglyceride (TG), total cholesterol (T-CHO), low-density lipoprotein cholesterol (LDL-C), high-density lipoprotein cholesterol (HDL-C), alanine aminotransferase (ALT), aspartate aminotransferase (AST) levels of serum were measured according to manufacturer’s protocol of each assay kit (Jiancheng, Nanjing, China).

For analysis of liver lipid content, 20 mg of frozen liver was homogenized in 10% (*w*/*v*) 50 mM Tris with 1% Triton X-100 using a hybrid grinding machine (Biheng Bio-Technique Co. Ltd., Shanghai, China). After 3000 rpm centrifugation for 1 min, the supernatants were measured according to manufacturer’s protocol of each assay kit (Jiancheng, Nanjing, China).

### 3.13. Histology and Microscopy

H&E staining was performed on formalin-fixed paraffin-embedded sections [47,48]. Brightfield images of liver were scanned with the NanoZoomer 2.0-HT slide scanner (Hamamatsu, Hamamatsu City, Japan) at 200× magnification. Brightfield images of eWAT were taken using MVX-TV1XC microscope (OLYMPUS, Tokyo, Japan) at 100× magnification.

### 3.14. Statistical Analysis

All data are shown as mean ± SEM. Statistical significance was determined using two-tailed Student’s *t*-test when comparing two groups. The *p* value less than 0.05 was considered statistically significant.

## 4. Discussion

In recent years, obesity and its complications have become a major global health issue, which threatens human health and affects economic stability worldwide [49]. At present, the underlying molecular mechanism of the pathogenesis and progression of obesity is not completely elucidated. However, increasing researches indicates that the gut microbiota dysbiosis is tightly associated with diet-induced obesity [50]. As a complex ecosystem in the human body, the intestinal flora maintains a dynamic balance with its host and plays an irreplaceable role in nutritional metabolism, host immunity and other vital activities. Gut microbes are an important bridge between diet and human health and play a vital role in maintaining the homeostasis in the human body [51]. Therefore, it is important to maintain the homeostasis of the human intestinal microbiota for the prevention and treatment of obesity.

Recently, accumulating data support that polysaccharides can improve obese condition by regulating intestinal microbiota [9,52,53]. Natural polysaccharides, as the most abundant biomolecules existing commonly in plants, animals and even algae, its chemical structure and biological function of natural polysaccharides are the third milestone in the search for the mysteries of life, after proteins and nucleic acids. There is growing evidence that CPPs are major and representative pharmacologically active macromolecules with a variety of biological activities in vitro and in vivo, such as immunomodulatory, antitumor, antioxidant, neuroprotective and antiviral [54] activities. These findings indicate that the CPP has potential for the treatment of obesity. The relationship between the anti-obesity effect of CPPs and the intestinal microbiota needs to be investigated, and the research on its mechanism has been further deepened.

## 5. Conclusions

In conclusion, a new homogeneous polysaccharide (CPP−1) was isolated from the root of *Codonopsis pilosula* (Franch.) Nannf., and its structure was characterized. CPP−1 is a neutral polysaccharide with molecular weight of 2.8 kDa, which was mainly composed of Man, Glc, Gal, and Ara at a molar ratio of 5.86 : 51.69 : 34.34 : 8.08. It was constituted by (1→)-linked-Glc residue, (1→3)-linked-Glc residues, (1→4)-linked-Gal residue, (1→2,3,4)-linked-Glc residue, (1→)-linked-Man residue, (1→3,4)-linked-Glc residue, and (1→)-linked-Ara residue. In this paper, we found for the first time that oral administration of CPP−1 can prevent obesity, as well as improve obesity-induced disorders of glucose metabolism, alleviate insulin resistance, and improve the effects of lipid metabolism. Our results confirmed the potential application of CPP−1 in the treatment of obesity. These results might be attributed to the change of gut microbial composition and regulation of lipid metabolism induced by prebiotic ability of polysaccharides. Therefore, *C. pilosula* could be a potential functional food source for preventing weight gain and obesity-related metabolic disorders.

## Figures and Tables

**Figure 1 molecules-27-05348-f001:**
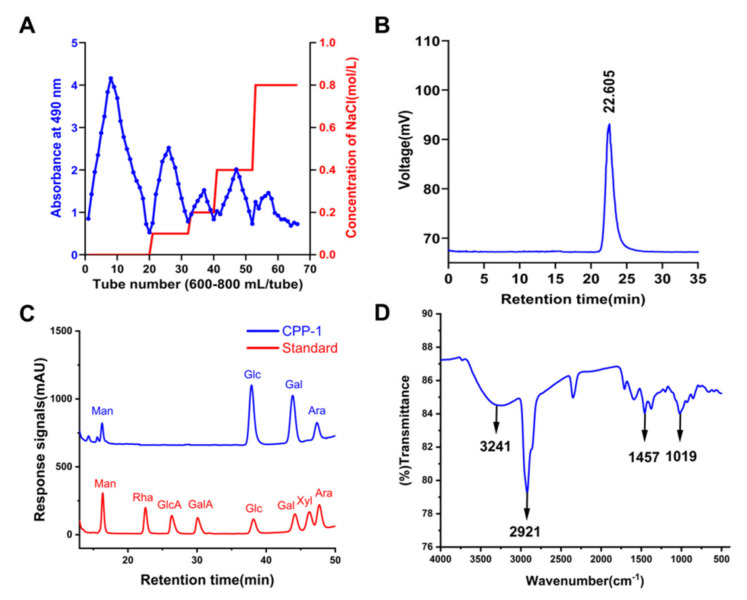
(**A**) The eluted profile of CPP80 on the DEAE-52 column; (**B**) HPGPC-ELSD result of CPP−1; (**C**) the results of monosaccharide composition analysis of CPP−1; (**D**) FT-IR spectrum of CPP−1 in the range of 4000–400 cm^−1^.

**Figure 2 molecules-27-05348-f002:**
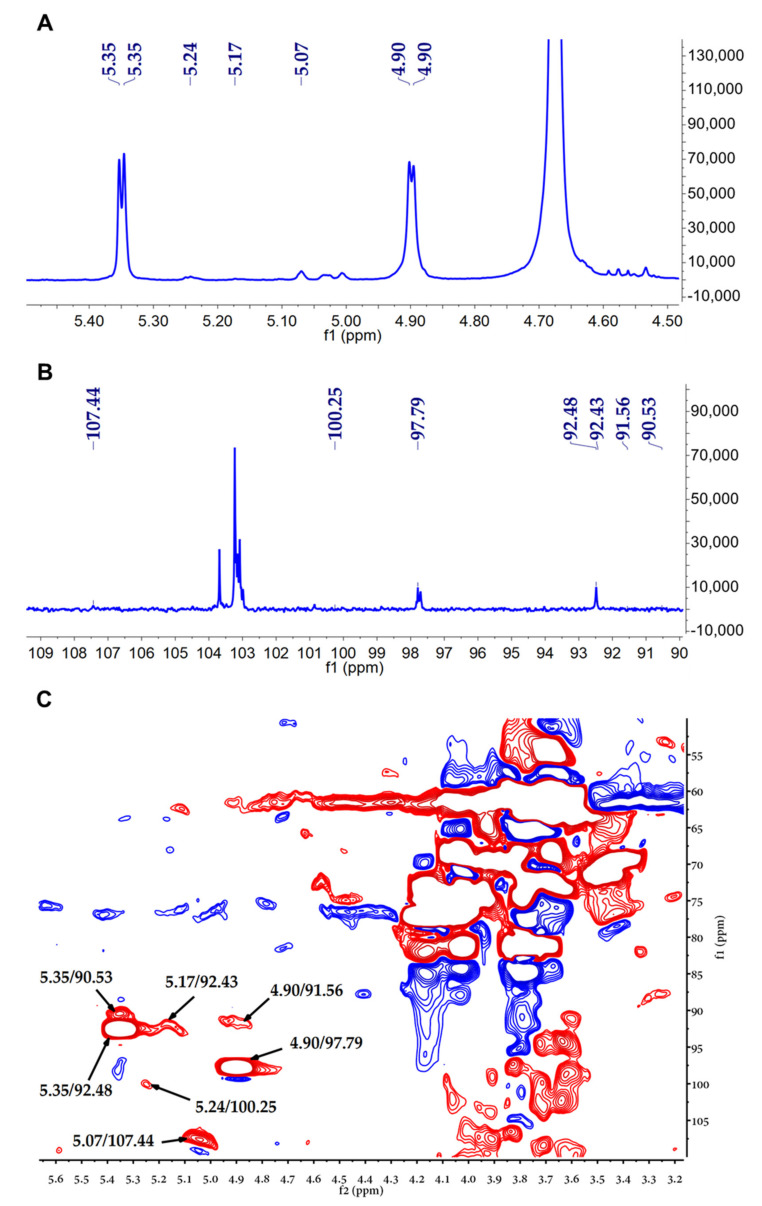
NMR spectra of CPP−1. (**A**) ^1^H NMR spectrum; (**B**) ^13^C NMR spectrum; (**C**) HSQC spectrum; red is a positive signal and blue is a negative signal.

**Figure 3 molecules-27-05348-f003:**
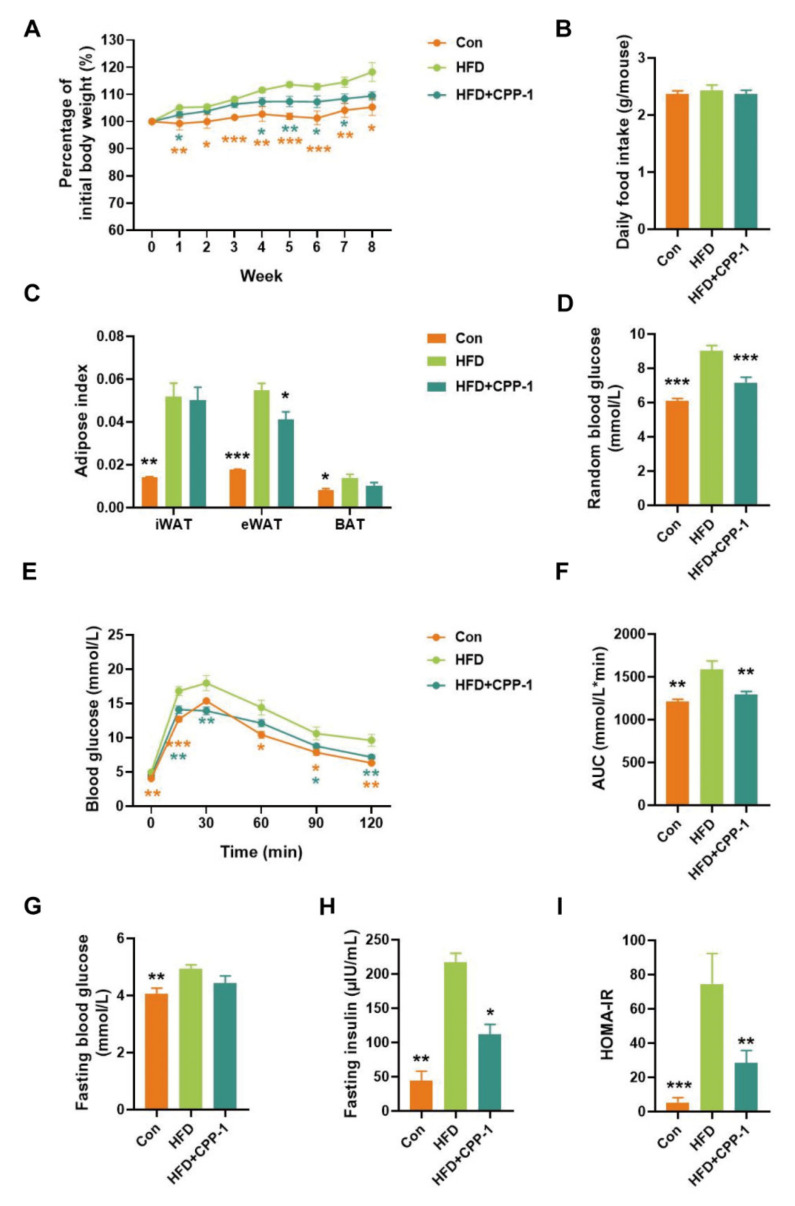
CPP−1 reduced HFD-induces body weight gain and improved glucose metabolism in mice. (**A**) Percentage of initial body weight; (**B**) daily food intake; (**C**) adipose index (adipose weight/body weight); (**D**) random blood glucose; (**E**) glucose tolerance test (GTT); (**F**) the areas under the curve (AUC); (**G**) fasting blood glucose; (**H**) fasting insulin; (**I**) insulin resistance index (HOMA-IR). Data are expressed as the mean ± SEM. Graph bars marked with different quantity of star sign on top represent statistically significant results based on two-tailed Student’s *t*-tests (** p* < 0.05, ** *p* < 0.01, **** p* < 0.001 versus HFD).

**Figure 4 molecules-27-05348-f004:**
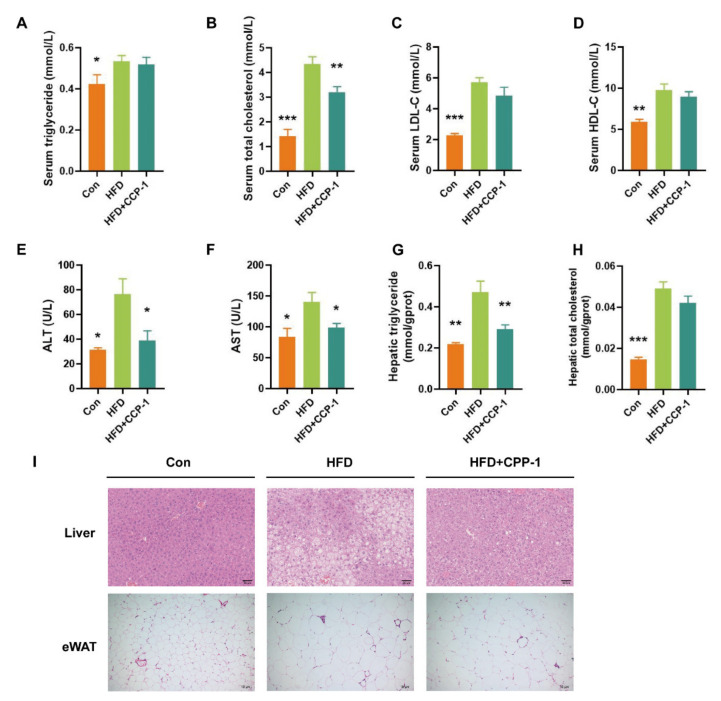
CPP−1 improves lipid metabolism and alleviates lipid accumulation in mice. (**A**) Serum TG; (**B**) serum T-CHO; (**C**) serum LDL-C; (**D**) serum HDL-C; (**E**) serum ALT; (**F**) serum AST; (**G**) hepatic TG; (**H**) hepatic T-CHO; (**I**) representative H&E staining of liver, scale bar: 50 μm (upper) and eWAT, scale bar: 10 μm (lower) sections. Data are expressed as the mean ± SEM (** p* < 0.05, *** p* < 0.01, **** p* < 0.001 versus HFD).

**Table 1 molecules-27-05348-t001:** Linkage patterns analysis of CPP−1.

Retention Time	Methylated	Type of Linkage	Major Mass Fragments (m/z)	Percentage(%)
8.739	2,3,4,6-Me_4_-Glc*p*	T-Glc*p*	43, 59, 102, 118, 129, 161, 205	20.53
9.856	2,4,6-Me_3_-Glc*p*	1,3-Glc*p*	43, 59, 87, 101, 118, 129, 161, 202, 234, 277	4.80
9.990	2,3,6-Me_3_-Gal*p*	1,4-Gal*p*	43, 59, 71, 87, 102, 118, 129, 162, 233	34.84
10.113	6-Me-Glc*p*	1,2,3,4-Glc*p*	87, 99, 115, 129, 157, 185, 218, 160, 333	10.36
10.698	2,3,4,6-Me_4_-Man*p*	T-Man*p*	43, 59, 102, 118, 129, 161, 205	5.17
10.933	2,6-Me_2_-Glc*p*	1,3,4-Glc*p*	43, 59, 87, 118, 129, 160, 185, 305	15.25
12.308	2,3,5-Me_3_-Ara*f*	T-Ara*f*	59, 71, 87, 102, 118, 129, 161, 162	9.05

## Data Availability

Data will be provided upon request.

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
