# Peer review of "The Obesity Amelioration Effect in High-Fat-Diet Fed Mice of a Homogeneous Polysaccharide from Codonopsis pilosula"

_molecules, 2022, doi:10.3390/molecules27165348_

Round 1
Reviewer 1 Report
The work by Su et al. describes the isolation and characterization of a polysaccharide CPP-1 isolated from Codonopsis pilosula root extracts. Moreover, the authors evaluated the effect of CPP-1 on body mass and glucose handling in mice fed with high fat diet.
In general terms, the study is well designed. The chemical characterization is nicely done, however, I have some major concerns with the experiments performed on mice.
1) There is an inconsistency in the protocol used in the glucose tolerance tests (GTT). In line 147, the authors mentioned that an intraperitoneal injection of glucose was used, but in the Methods section is was stated that glucose was administered by gavage for GTT. Please correct the inconsistency.
2) The graph shown in Figure 4A shows the fasting glucose levels, however, the authors stated in the Results section that glucose levels shown in graph 4A correspond to glucose levels after glucose injection.
In the case that graph shown in Figure 4A corresponds to glucose levels after glucose administration, why the glucose levels are different than the levels shown in the first points in Figure 4B?
I recommend the acceptance of this manuscript after addressing the concerns raised above.
Author Response
Our responses to Reviewer 1 comments:
The work by Su et al. describes the isolation and characterization of a polysaccharide CPP-1 isolated from Codonopsis pilosula root extracts. Moreover, the authors evaluated the effect of CPP-1 on body mass and glucose handling in mice fed with high fat diet.
In general terms, the study is well designed. The chemical characterization is nicely done, however, I have some major concerns with the experiments performed on mice.
- There is an inconsistency in the protocol used in the glucose tolerance tests (GTT). In line 147, the authors mentioned that an intraperitoneal injection of glucose was used, but in the Methods section is was stated that glucose was administered by gavage for GTT. Please correct the inconsistency.
Response: Sorry for mistakes. The method of GTT performed in the article is orally gavage glucose. In the article "Materials and Methods", the information in 3.10 and Figure 3E has been modified as follows.
3.10. Glucose Tolerance Test
The mice were fasted for 16 h and oral gavage of D-glucose (2 g/kg body weight). Blood glucose levels at 0, 15, 30, 60, 90 and 120 min post glucose administration were measured with blood samples taken from tail veins using blood sugar test paper and glucometer (Sannuo Biosensors, China).(Line 277-281)
- The graph shown in Figure 4A shows the fasting glucose levels, however, the authors stated in the Results section that glucose levels shown in graph 4A correspond to glucose levels after glucose injection.
Response: Thank you for pointing out this mistake. The biochemical indices of each group have been rearranged, and we have checked the graphs and the corresponding result descriptions as follows.
2.7 CPP-1 Supplementation Improves Blood Glucose Metabolism
Extensive evidence has shown that obesity is associated with abnormal glucose tolerance [42]. CPP-1 treatment significantly lowered random blood glucose in HFD-fed mice (Fig.3D), indicating the improved glucose metabolism effect of CPP-1 supplementation. To further investigate the effect of CPP-1 on obesity-related glucose metabolism dysfunction, the glucose tolerance test (GTT) was performed in three groups of mice. As shown in Fig. 3E, after orally gavage glucose, the HFD group dis-played a remarkably higher level of blood glucose than the control group, while the impaired glucose tolerance was reversed by CPP-1 supplementation. (Line 148-155)
- In the case that graph shown in Figure 4A corresponds to glucose levels after glucose administration, why the glucose levels are different than the levels shown in the first points in Figure 4B?
Response: Thanks a lot for your kind comments. We have supplemented the data of biochemical indicators and rearranged them. The graph shown in Figure.3G shows the fasting glucose levels, fasting blood glucose is the blood glucose value measured by the plasma taken before breakfast after an overnight fast (at least 8 to 10 hours without any food, except water). Figure.3E shows curve of impaired glucose tolerance tests, which is defined as the blood glucose value detected in plasma after 16h of fasting. These two experiments have different time periods and fasting times, so the glucose levels start at different points.
I recommend the acceptance of this manuscript after addressing the concerns raised above.

Reviewer 2 Report
Overall the introduction section is below the standard and not well addresses the problem. Besides, the provided background study is not sufficient. Hence this section needs major corrections.
In the first paragraph, “Obesity has been identified as an epidemic and is a major public health problem prevalent worldwide [1]. Obesity can cause or exacerbate many health problems, such as metabolic syndrome, cardiovascular disease, and sleep and breathing disorders [2-4]. It is therefore important to accelerate the prevention and treatment of obesity. At present, dietary intervention and pharmacotherapy are common choices for long-term weight reduction, but they are usually compromised mainly due to poor compliance and side effects [5,6]. On the contrary, it is worth noting that polysaccharides effectively prevent or treat obesity with few side effects [7]”.
Line 26-27: The authors start both the sentences with the same word “Obesity” to avoid repetition of the words.
Please extend this paragraph, by citing some relevant and latest citations.
Similarly, in the second paragraph, “Polysaccharides, as biological macromolecules existing widely in nature, have a variety of physiological functions and biological activities. It has been demonstrated that polysaccharides from Traditional Chinese medicine, such as Lycium barbarum[11], Morus alba L.[12], Ganoderma amboinense[13], and Chinese yam[14], can prevent or suppress obesity. Considering the well-recognized anti-obesity potentials of polysaccharides, Codonopsis pilosula polysaccharides (CPS) may serve as another class of natural anti-obesity agents. Studies have shown that CPS contributes to the improvement of metabolic syndrome. For example, CPS significantly lowered blood glucose and significantly improved lipid metabolism disorders in diabetic rats. Obesity, like diabetes, is a metabolic syndrome. As far as we know, studies on the biological activity of purified CPS, especially in anti-obesity have been rarely explored.”
Is about the anti-obesity potential of polysaccharides, but the authors did provide any citation, especially in lines 38-40. Please modify this paragraph to support and justify the title.
Overall, the results are presented well. The provided figures and tables well support the results. However, some grammatical mistakes are found that need improvement. Secondly, please recheck the p-values mentioned in figure 3.
However, the discussion section is not sufficient please discuss your results with the previously published literature.
All the figures must be cited in the text, for example, fig-3c?
Please add recommendations at the end of the conclusion section.
Author Response
Our responses to Reviewer 2 comments:
Overall the introduction section is below the standard and not well addresses the problem. Besides, the provided background study is not sufficient. Hence this section needs major corrections.
- In the first paragraph, “Obesity has been identified as an epidemic and is a major public health problem prevalent worldwide [1]. Obesity can cause or exacerbate many health problems, such as metabolic syndrome, cardiovascular disease, and sleep and breathing disorders [2-4]. It is therefore important to accelerate the prevention and treatment of obesity. At present, dietary intervention and pharmacotherapy are common choices for long-term weight reduction, but they are usually compromised mainly due to poor compliance and side effects [5,6]. On the contrary, it is worth noting that polysaccharides effectively prevent or treat obesity with few side effects [7]”.
- Line 26-27: The authors start both the sentences with the same word “Obesity” to avoid repetition of the words.
Please extend this paragraph, by citing some relevant and latest citations.
Response: Thanks a lot for your kind comments. We have avoided repetition of words and replaced citations in the article with literature from the last six years, as follows.
Obesity is an epidemic and a major public health problem prevalent worldwide[1], which can cause or exacerbate many health problems, such as metabolic syndrome, cardiovascular disease, sleep and breathing disorders[2-4]. It is therefore important to accelerate the prevention and treatment of obesity. At present, dietary intervention and pharmacotherapy are common choices for long-term weight reduction, but they are usually largely compromised mainly due to poor compliance and side effects[5,6]. On the contrary, it is worth noting that polysaccharides effectively prevent or treat obesity with few side effects[7-9].(Line 27-34)
Ref:
- Endalifer, M.L.; Diress, G. Epidemiology, Predisposing Factors, Biomarkers, and Prevention Mechanism of Obesity: A Systematic Review. J. Obesity. 2020, 2020, 6134362.
- Jokinen, E. Obesity and cardiovascular disease. Pediatr. 2015, 67, 25–32.
- Han, T.S.; Lean, M.E. A clinical perspective of obesity, metabolic syndrome and cardiovascular disease. Cardiovasc. Dis. 2016, 5, 1–13.
- Almendros, I.; Martinez-Garcia, M.A.; Farré, R.; Gozal, D. Obesity, sleep apnea, and cancer. J. Obesity. 2020, 44, 1653–1667.
- Vink, R.G.; Roumans, N.J.T.; Arkenbosch, L.A.J.; Mariman, E.C.M.; van Baak, M.A. The effect of rate of weight loss on long‐term weight regain in adults with overweight and obesity. 2016, 24, 321–327.
- Bessesen, D.H.; Van Gaal, L.F. Progress and challenges in anti-obesity pharmacotherapy. Diabetes. Endocrinol. 2018, 6, 237–248.
- Su, Y.; Li, J.; Wu, L.; Kuang, H. Polysaccharides from TCM herbs exhibit potent anti-obesity effect by mediating the community structure of gut microbiota. Die Pharmazie-An International Journal of Pharmaceutical Sciences 2021, 76, 473–479.
- Zhang, Y.; Xie, Q.; You, L.; Cheung, P.C.K.; Zhao, Z. Behavior of Non-Digestible Polysaccharides in Gastrointestinal Tract: A Mechanistic Review of its Anti-Obesity Effect. 2021, 2, 59–72.
- Lee, H.B.; Kim, Y.S.; Park, H.Y. Pectic polysaccharides: Targeting gut microbiota in obesity and intestinal health. Polym. 2022, 119363.
- Similarly, in the second paragraph, “Polysaccharides, as biological macromolecules existing widely in nature, have a variety of physiological functions and biological activities. It has been demonstrated that polysaccharides from Traditional Chinese medicine, such as Lycium barbarum[11], Morus alba L.[12], Ganoderma amboinense[13], and Chinese yam[14], can prevent or suppress obesity. Considering the well-recognized anti-obesity potentials of polysaccharides, Codonopsis pilosula polysaccharides (CPS) may serve as another class of natural anti-obesity agents. Studies have shown that CPS contributes to the improvement of metabolic syndrome. For example, CPS significantly lowered blood glucose and significantly improved lipid metabolism disorders in diabetic rats. Obesity, like diabetes, is a metabolic syndrome. As far as we know, studies on the biological activity of purified CPS, especially in anti-obesity have been rarely explored.”
Is about the anti-obesity potential of polysaccharides, but the authors did provide any citation, especially in lines 38-40. Please modify this paragraph to support and justify the title.
Response: Thanks a lot for your kind comments. We have added citations to the article to support and justify the article as follows:
Polysaccharides, as biological macromolecules existing widely in nature, have a variety of physiological functions and biological activities[10]. It has been demonstrat-ed that polysaccharides from Traditional Chinese medicine, such as Lycium barbarum[11], Morus alba L.[12], Ganoderma amboinense[13], and Dioscorea opposite[14], can prevent or suppress obesity. Considering the well-recognized anti-obesity potentials of polysaccharides[8], Codonopsis pilosula polysaccharides (CPS) may serve as another class of natural anti-obesity agents[15]. Studies have shown that CPS contribute to the im-provement of metabolic syndrome[16-18]. For example, CPS significantly lowered blood glucose[16] and significantly improved lipid metabolism disorders in diabetic rats[19]. Obesity, like diabetes, is a metabolic syndrome[20]. As far as we know, studies on the biological activity of purified CPS, especially in anti-obesity has been rarely explored.(Line35-45)
Ref:
- Xie, J.H.; Jin, M.L.; Morris, G.A.; Zha, X.Q.; Chen, H.Q.; Yi, Y.; Li, J.E.; Wang, Z.-J.; Gao, J.; Nie, S.-P. Advances on bioactive polysaccharides from medicinal plants. Rev. Food. Sci. 2016, 56, S60–S84.
- Yang, Y.; Chang, Y.; Wu, Y.; Liu, H.; Liu, Q.; Kang, Z.; Wu, M.; Yin, H.; Duan, J. A homogeneous polysaccharide from Lycium barbarum: Structural characterizations, anti-obesity effects and impacts on gut microbiota. J. Biol. Macromol. 2021, 183, 2074–2087.
- Li, R.; Xue, Z.; Jia, Y.; Wang, Y.; Li, S.; Zhou, J.; Liu, J.; Zhang, M.; He, C.; Chen, H. Polysaccharides from mulberry (Morus alba) leaf prevents obesity by inhibiting pancreatic lipase in high-fat diet induced mice. Int. J. Biol. Macromol. 2021, 192, 452–460.
- Ren, F.; Meng, C.; Chen, W.; Chen, H.; Chen, W. Ganoderma amboinense polysaccharide prevents obesity by regulating gut microbiota in high-fat-diet mice. Biosci. 2021, 42, 101107.
- Cheng, Z.; Hu, M.; Tao, J.; Yang, H.; Yan, P.; An, G.; Wang, H. The protective effects of Chinese yam polysaccharide against obesity-induced insulin resistance. Funct. Foods. 2019, 55, 238–247.
- Bai, R.; Cui, F.; Li, W.; Wang, Y.; Wang, Z.; Gao, Y.; Wang, N.; Xu, Q.; Hu, F.; Zhang, Y. Codonopsis pilosula oligosaccharides modulate the gut microbiota and change serum metabolomic profiles in high fat diet induced obese mice. Funct. 2022.
- Liu, W.; Lv, X.; Huang, W.; Yao, W.; Gao, X. Characterization and hypoglycemic effect of a neutral polysaccharide extracted from the residue of Codonopsis Pilosula. Polym. 2018, 197, 215–226.
- Hu, Y.R.; Xing, S.L.; Chen, C.; Shen, D.Z.; Chen, J.L. Codonopsis pilosula polysaccharides alleviate Aβ1-40-induced PC12 cells energy dysmetabolism via CD38/NAD+ signaling pathway. Alzheimer. Res. 2021, 18, 208–221.
- Li, J.; Zhang, X.; Cao, L.; Ji, J.; Gao, J. Three inulin-type fructans from Codonopsis pilosula (Franch.) Nannf. Roots and their prebiotic activity on Bifidobacterium longum. 2018, 23, 3123.
- Cao, L.; Du, C.; Zhai, X.; Li, J.; Meng, J.; Shao, Y.; Gao, J. Codonopsis pilosula Polysaccharide Improved Spleen Deficiency in Mice by Modulating Gut Microbiota and Energy Related Metabolisms. Pharmacol. 2022, 1399.
- Engin, A. The definition and prevalence of obesity and metabolic syndrome. Obesity and lipotoxicity 2017, 1–17.
- Overall, the results are presented well. The provided figures and tables well support the results. However, some grammatical mistakes are found that need improvement. Secondly, please recheck the p-values mentioned in figure 3.
Response: Thank you very much for you appreciation. We used two-tailed Student t-test to determine whether the data from the two groups (Con group vs HFD group; HFD group vs HFD + CPP-1 group) were statistically different. We have revised Fig.3A and 3B as follows.
Figure 3. CPP-1 reduced HFD-induces body weight gain and improved glucose metabolism in mice. (A) Percentage of initial body weight; (B) Daily food intake; (C) Adipose index (Adipose weight/body weight); (D) Random blood glucose; (E) Glucose tolerance test (GTT) and areas under the curve (AUC); (F) Fasting blood glucose; (G) Fasting insulin; (H) Insulin resistance index (HOMA-IR). Data are expressed as the mean ± SEM. Graph bars marked with different quantity of star sign on top represent statistically significant results based on two-tailed Student’s t-tests (*p < 0.05, **p < 0.01, ***p < 0.001 versus HFD).
- However, the discussion section is not sufficient please discuss your results with the previously published literature.
Response: We thank the reviewer for this insightful comment. We discuss the results of this study with the previously published literature as follows.
- Discussion
In recent years, obesity and its complications have become a major global health issue, which threatens human health and affects economic stability worldwide[49]. At present, the underlying molecular mechanism of the pathogenesis and progression of obesity is not completely elucidated. However, increasing researches indicates that the gut microbiota dysbiosis is tightly associated with diet-induced obesity[50]. As a com-plex ecosystem in the human body, the intestinal flora maintains a dynamic balance with its host and plays an irreplaceable role in nutritional metabolism, host immunity and other vital activities. Gut microbes are an important bridge between diet and hu-man health and play a vital role in maintaining the homeostasis in the human body[51]. Therefore, it is important to maintain the homeostasis of the human intestinal micro-biota for the prevention and treatment of obesity.
Recently, accumulating data support that polysaccharides can improve obese condition by regulating intestinal microbiota[52-54]. Natural polysaccharides, as the most abundant biomolecules existing commonly in plants, animals and even algae, its chemical structure and biological function of natural polysaccharides are the third milestone in the search for the mysteries of life, after proteins and nucleic acids. There is growing evidence that CPPs are major and representative pharmacologically active macromolecules with a variety of biological activities in vitro and in vivo, such as immunomodulatory, antitumor, antioxidant, neuroprotective and antiviral[55] activities. These findings indicates that the CPP has potential for the treatment of obesity. The relationship between the anti-obesity effect of CPPs and the intestinal microbiota needs to be investigated, and the research on its mechanism has been further deepened.
Ref:
- Ansari, S.; Haboubi, H.; Haboubi, N. Adult obesity complications: challenges and clinical impact. Adv. Endocrinol. 2020, 11.
- Song, X.; Wang, L.; Liu, Y.; Zhang, X.; Weng, P.; Liu, L.; Zhang, R.; Wu, Z. The gut microbiota–brain axis: Role of the gut microbial metabolites of dietary food in obesity. Res. Int. 2022, 153, 110971.
- Uchiyama, K.; Naito, Y.; Takagi, T. Intestinal microbiome as a novel therapeutic target for local and systemic inflammation. Therapeut. 2019, 199, 164–172.
- Huang, Y.; Chen, H.; Zhang, K.; Lu, Y.; Wu, Q.; Chen, J.; Li, Y.; Wu, Q.; Chen, Y. Extraction, purification, structural characterization, and gut microbiota relationship of polysaccharides: A review. J. Biol. Macromol. 2022, 213, 967–986.
- Lee, H.B.; Kim, Y.S.; Park, H.Y. Pectic polysaccharides: Targeting gut microbiota in obesity and intestinal health. Polym. 2022, 287, 119363.
- Fang, Q.; Hu, J.; Nie, Q.; Nie, S. Effects of polysaccharides on glycometabolism based on gut microbiota alteration. Food. Sci. Tech. 2019, 92, 65–70.
- Luan, F.; Ji, Y.; Peng, L.; Liu, Q.; Cao, H.; Yang, Y.; He, X.; Zeng, N. Extraction, purification, structural characteristics and biological properties of the polysaccharides from Codonopsis pilosula: A review. Polym. 2021, 261, 117863.
- All the figures must be cited in the text, for example, fig-3c?
Response: Those comments are very helpful for us. The biochemical indices of each group have been rearranged. All the figures have been cited in the text.
- Please add recommendations at the end of the conclusion section.
Response: Thank you for your comments concerning our manuscript. Recommendations and outlooks have been added to the results.
These results might be attributed to the change of gut microbial composition and regulation of lipid metabolism induced by prebiotic ability of polysaccharides. Therefore, C. pilosula could be a potential functional food source for preventing weight gain and obesity-related metabolic disorders. (Line 317-321)

Reviewer 3 Report
The authors reported the anti-obesity effect of polysaccharide from Codonopsis pilosula (CPP-1). The manuscript is needed to be revised. For example:
1. Authors claim that CPP-1 can potentially ameliorate obesity through high-fat consumption. The authors showed the effect of this substance on body weight and blood glucose; however, these parameters are insufficient to indicate the effectiveness of CPP-1 on obesity.
2. The authors failed to discuss their findings. In the results and discussion section, the authors should begin with a result and then discuss the significance and implications of their findings. Authors should compare with other studies.
3. The format of reference on lines 183-184.
4. Line 192 The sample for HPLC is CPP-1 or CPP80-1?
Author Response
Our responses to Reviewer 3 comments:
The authors reported the anti-obesity effect of polysaccharide from Codonopsis pilosula (CPP-1). The manuscript is needed to be revised. For example:
- Authors claim that CPP-1 can potentially ameliorate obesity through high-fat consumption. The authors showed the effect of this substance on body weight and blood glucose; however, these parameters are insufficient to indicate the effectiveness of CPP-1 on obesity.
Response: Thanks a lot for your kind comments. In the revised manuscript, we provide more proofs about anti-obesity effect of CPP-1, including improved glucose and lipid metabolism and improved morphology of liver and adipose tissues. The figures have been shown in Fig.3D-H and Fig.4A-D, G-I as follows.
Figure 3. CPP-1 reduced HFD-induces body weight gain and improved glucose metabolism in mice. (A) Percentage of initial body weight; (B) Daily food intake; (C) Adipose index (Adipose weight/body weight); (D) Random blood glucose; (E) Glucose tolerance test (GTT) and areas under the curve (AUC); (F) Fasting blood glucose; (G) Fasting insulin; (H) Insulin resistance index (HOMA-IR). Data are expressed as the mean ± SEM. Graph bars marked with different quantity of star sign on top represent statistically significant results based on two-tailed Student’s t-tests (*p < 0.05, **p < 0.01, ***p < 0.001 versus HFD).
Figure 4. CPP-1 improves lipid metabolism and alleviates lipid accumulation in mice. (A) Serum TG; (B) Serum T-CHO; (C) Serum LDL-C; (D) Serum HDL-C; (E) Serum ALT; (F) Serum AST; (G) Hepatic TG; (H) Hepatic T-CHO; (I) Representative H&E staining of liver (upper) and eWAT (lower) sections. Data are expressed as the mean ± SEM (*p < 0.05, **p < 0.01, ***p < 0.001 versus HFD).
- The authors failed to discuss their findings. In the results and discussion section, the authors should begin with a result and then discuss the significance and implications of their findings. Authors should compare with other studies.
Response: Thank you very much for you appreciation. The results of this article have been discussed in the context of the article as follows.
- Discussion
In recent years, obesity and its complications have become a major global health issue, which threatens human health and affects economic stability worldwide[49]. At present, the underlying molecular mechanism of the pathogenesis and progression of obesity is not completely elucidated. However, increasing researches indicates that the gut microbiota dysbiosis is tightly associated with diet-induced obesity[50]. As a complex ecosystem in the human body, the intestinal flora maintains a dynamic balance with its host and plays an irreplaceable role in nutritional metabolism, host immunity and other vital activities. Gut microbes are an important bridge between diet and hu-man health and play a vital role in maintaining the homeostasis in the human body[51]. Therefore, it is important to maintain the homeostasis of the human intestinal microbiota for the prevention and treatment of obesity.
Recently, accumulating data support that polysaccharides can improve obese condition by regulating intestinal microbiota[52-54]. Natural polysaccharides, as the most abundant biomolecules existing commonly in plants, animals and even algae, its chemical structure and biological function of natural polysaccharides are the third milestone in the search for the mysteries of life, after proteins and nucleic acids. There is growing evidence that CPPs are major and representative pharmacologically active macromolecules with a variety of biological activities in vitro and in vivo, such as immunomodulatory, antitumor, antioxidant, neuroprotective and antiviral[55] activities.
These findings indicates that the CPP-1 has potential for the treatment of obesity. The relationship between the anti-obesity effect of CPPs and the intestinal microbiota needs to be investigated, and the research on its mechanism has been further deepened.
- Conclusions
In conclusion, a novel homogeneous polysaccharide (CPP-1) was isolated from the root of Codonopsis pilosula, and its structure was characterized. CPP-1 is a neutral polysaccharide with molecular weight of 2.8 kDa, which was mainly composed of Man, Glc, Gal and Ara at a molar ratio of 5.86: 51.69 : 34.34: 8.08. It was constituted by (1→4)-linked-Gal residue, (1→3)-linked-Glc residues, (1→3,4)-linked-Glc residue, (1→2,3,4)-linked-Glc residue, (1→)-linked-Glc residue, (1→)-linked-Man residue and (1→)-linked-Ara residue. In this paper, we found for the first time that oral administration of CPP-1 can prevent obesity, as well as improve obesity-induced disorders of glucose metabolism, alleviate insulin resistance and improve the effects of lipid metabolism. Our results confirmed the potential application of CPP-1 in the treatment of obesity. These results might be attributed to the change of gut microbial composition and regulation of lipid metabolism induced by prebiotic ability of polysaccharides. Therefore, C. pilosula could be a potential functional food source for preventing weight gain and obesity-related metabolic disorders.
- The format of reference on lines 183-184.
Response: Sorry for mistakes. We have modified the reference format as required.
- Line 192 The sample for HPLC is CPP-1 or CPP80-1?
Response: Thank for this point. The sample for HPLC is CPP-1 as follows.
Figure 1. (A) The eluted profile of CPP80 on the DEAE-52 column; (B) HPGPC-ELSD result of CPP-1; (C) The results of monosaccharide composition analysis of CPP-1; (D) FT-IR spectrum of CPP-1 in the range of 4000-400 cm-1.
By eluting through a Sephacryl S-200 HR gel column (2.5 cm × 100 cm), a homogenous polysaccharide CPP-1 was finally obtained. As shown in Fig. 1B, CPP-1 showed a symmetrical peak, indicating it was a homogeneous polysaccharide and the molecular weight was 2.8 kDa according to the standard curve (lgMp = -0.3457tR + 11.084). ( line 80-84)

Reviewer 4 Report
This is a very interesting manuscript which urges us to look closer into the potential of this natural product and realize its full potential.
I invite the authors to consider the following suggestions:
Line 33-36: Kindly revisit the references as No. 10 seems to be missing.
Line 38-44: Please include relevant references for these statements. This is a long segment to be without any reference to support it. “Studies have shown that CPS contribute to the improvement of metabolic syndrome.”, “CPS significantly lowered blood glucose and significantly improved lipid metabolism disorders in diabetic rats.”, and “Obesity, like diabetes, is a metabolic syndrome.” Would require some key studies that reflect these findings and hence support the statements.
Line 43-44: Kindly consider rephrasing.
Line 46-48: Please include some key references here as well. It is essential for the reader to understand that this natural product has been utilized in TCM long before its full potential was realized via the scientific method, hence emphasize on how crucial folk remedies are for the identification of new agents.
Line 146-152: Kindly consider including the relevant values of the changes observed in the text as well (apart for them being presented in the graph). Since this work can be used as reference for future research on this topic it is essential for the reader to have a clear view of the outcomes in order to draw comparisons.
Discussion: Perhaps the authors would consider keeping the format of the manuscript as provided by the journal and maintain a separate segment for the discussion part. As it is (Results and Discussion) there seems to be no discussion of this study’s outcomes related to previous research in any of the assays performed. Kindly revisit as it may also help the authors to construct a more coherent conclusion segment as well.
Author Response
Our responses (Red words) to Reviewer 4 comments:
This is a very interesting manuscript which urges us to look closer into the potential of this natural product and realize its full potential. I invite the authors to consider the following suggestions:
- Line 33-36: Kindly revisit the references as No. 10 seems to be missing.
Response: Thank you very much for you appreciation. The references No. 10 has been added to the article in line 36. We have added and organized the references in the article as follows.
Polysaccharides, as biological macromolecules existing widely in nature, have a variety of physiological functions and biological activities[10]. (line 35-36)
- Line 38-44: Please include relevant references for these statements. This is a long segment to be without any reference to support it. “Studies have shown that CPS contribute to the improvement of metabolic syndrome.”, “CPS significantly lowered blood glucose and significantly improved lipid metabolism disorders in diabetic rats.”, and “Obesity, like diabetes, is a metabolic syndrome.” Would require some key studies that reflect these findings and hence support the statements.
- Line 43-44: Kindly consider rephrasing.
Response: Those comments are very helpful for us. Relevant references have been provided to support these statements as follows.
Studies have shown that CPS contribute to the improvement of metabolic syndrome[16-18]. For example, CPS significantly lowered blood glucose[16] and significantly improved lipid metabolism disorders in diabetic rats[19]. Obesity, like diabetes, is a metabolic syndrome[20]. As far as we know, studies on the biological activity of purified CPS, especially in anti-obesity has been rarely explored.(lines 41-45)
Ref:
- Liu, W.; Lv, X.; Huang, W.; Yao, W.; Gao, X. Characterization and hypoglycemic effect of a neutral polysaccharide extracted from the residue of Codonopsis Pilosula. Polym. 2018, 197, 215–226.
- Hu, Y.R.; Xing, S.L.; Chen, C.; Shen, D.Z.; Chen, J.L. Codonopsis pilosula polysaccharides alleviate Aβ1-40-induced PC12 cells energy dysmetabolism via CD38/NAD+ signaling pathway. Alzheimer. Res. 2021, 18, 208–221.
- Li, J.; Zhang, X.; Cao, L.; Ji, J.; Gao, J. Three inulin-type fructans from Codonopsis pilosula (Franch.) Nannf. Roots and their prebiotic activity on Bifidobacterium longum. Molecules. 2018,23, 3123.
- Cao, L.; Du, C.; Zhai, X.; Li, J.; Meng, J.; Shao, Y.; Gao, J. Codonopsis pilosula Polysaccharide Improved Spleen Deficiency in Mice by Modulating Gut Microbiota and Energy Related Metabolisms. Pharmacol. 2022, 13, 862763.
- Engin, A. The definition and prevalence of obesity and metabolic syndrome. Obesity and lipotoxicity. 2017, 1–17.
- Line 46-48: Please include some key references here as well. It is essential for the reader to understand that this natural product has been utilized in TCM long before its full potential was realized via the scientific method, hence emphasize on how crucial folk remedies are for the identification of new agents.
Response: Thank you very much for you appreciation. Relevant references have been provided to support this statement as follows.
Radix Codonopsis is the dried root of Codonopsis pilosula (Franch.) Nannf., C. pilosula Nannf. var. modesta (Nannf.) L. T. Shen or C.tangshen Oliv. Radix Codonopsis has a wide range of branches in China and has a long history of medicinal use. Of the above three kinds of Radix Codonopsis, C. pilosula is the most widely used in clinical practice[21].(Line 46-48)
- Gao, S.M.; Liu, J.S.; Wang, M.; Cao, T.T.; Qi, Y.D.; Zhang, B.G.; Sun, X.B.; Liu, H.T.; Xiao, P.G. Traditional uses, phytochemistry, pharmacology and toxicology of Codonopsis: A review. Ethnopharmacol. 2018, 219, 50–70.
- Line 146-152: Kindly consider including the relevant values of the changes observed in the text as well (apart for them being presented in the graph). Since this work can be used as reference for future research on this topic it is essential for the reader to have a clear view of the outcomes in order to draw comparisons.
Response: Thank you very much to the reviewers. In the revised manuscript, we have discussed the difference of fasting blood glucose in three groups. This part has been added in Line 160-166 as follows.
The levels of fasting insulin showed significantly different among three groups and the HFD+CPP-1 group exhibited a significantly lower fasting insulin level (Fig.3G), indicating ameliorating effects on insulin resistance in HFD+CPP-1 group. However, in the HFD group, the mean HOMA-IR index is as high as 74.5, indicating severe insulin resistance. Compared to the HFD group, the HFD+CPP-1 group displayed lower HOMA-IR index (mean value = 28.4), confirming the insulin improvement effect of CPP-1 treatment (Fig.3H).
- Discussion: Perhaps the authors would consider keeping the format of the manuscript as provided by the journal and maintain a separate segment for the discussion part. As it is (Results and Discussion) there seems to be no discussion of this study’s outcomes related to previous research in any of the assays performed. Kindly revisit as it may also help the authors to construct a more coherent conclusion segment as well.
Response: Those comments are very helpful for us. We have added separate segment for the discussion part as follows.(line 322-344)
- Discussion
In recent years, obesity and its complications have become a major global health issue, which threatens human health and affects economic stability worldwide[49]. At present, the underlying molecular mechanism of the pathogenesis and progression of obesity is not completely elucidated. However, increasing researches indicates that the gut microbiota dysbiosis is tightly associated with diet-induced obesity[50]. As a complex ecosystem in the human body, the intestinal flora maintains a dynamic balance with its host and plays an irreplaceable role in nutritional metabolism, host immunity and other vital activities. Gut microbes are an important bridge between diet and human health and play a vital role in maintaining the homeostasis in the human body[51]. Therefore, it is important to maintain the homeostasis of the human intestinal microbiota for the prevention and treatment of obesity.
Recently, accumulating data support that polysaccharides can improve obese condition by regulating intestinal microbiota[52-54]. Natural polysaccharides, as the most abundant biomolecules existing commonly in plants, animals and even algae, its chemical structure and biological function of natural polysaccharides are the third milestone in the search for the mysteries of life, after proteins and nucleic acids. There is growing evidence that CPPs are major and representative pharmacologically active macromolecules with a variety of biological activities in vitro and in vivo, such as immunomodulatory, antitumor, antioxidant, neuroprotective and antiviral[55]. These findings indicates that the CPP has potential for the treatment of obesity. The relationship between the anti-obesity effect of CPPs and the intestinal microbiota needs to be investigated, and the research on its mechanism has been further deepened.

Reviewer 5 Report
Su Q. et al investigated the anti-obesity from natural resources like the root of Codonopsis pilosula which reflects well multi-physiological functions such as anti-inflammatory activity, anti-tumor activity, and anti-diabetic activity, hypoglycemic, Anti-viral, antioxidant, and prebiotic in animal models. In this report, they challenge to extract bioactive compounds and isolate them and characterize them by utilizing physical-chemical methodologies. To determine anti-obesity activity, this novel compound was examined in its biological functions using HFD-induced mice obesity followed by treatment of named CPP-1 novel compound under their report entitled “
The obesity amelioration effect in high-fat-diet fed mice of a homogeneous polysaccharide from Codonopsis pilosula.” It is very interesting and neatly documented. However, several points need to be confirmed and validated by promoting going through Q/A sessions as follows.
The authors mentioned they used n=10 mice without sex description in detail. So, I Was wondering about gender differences, and would be helpful if they describe it
It would be great if they can provide a complete structure of CPP-1
Regarding hyperglycemic measurement, are there any parameters available to demonstrate the CPP-1 ameliorate fat tissue or systemic level? Is any histological data available to show CPP-1 as an anti-obesogenic effect in the C57Bl/6 mice model? For example, is there any difference in H&E staining using epididymal adipose tissues with serum biochemical analysis (e.g., TG, TC, plasma glucose, plasma insulin, HDL/LDL, etc)?
What do they believe is the molecular target of CPP-1if it is supposed to be true CPP-1 is the candidate of anti-obesogenic compounds? Is there any evidence?
Have authors measured any PKJ/PD value of CPP-1 in blood levels or have been measured half -of a lifetime (T2/1) of CPP-1 in blood?
Is this CPP-1 circulating to the brain-blood barrier or not?
Table 1 methylation data, have they measured retention time depend on their detection condition which may need to be described in detail in the 3.6 at M & M.
Have they confirmed it by TLC analysis to confirm the sugar moiety of CPP-1
In the title, the authors use homogeneous polysaccharides… is there any criteria for homogeneity in the final structure of CPP-1?
Limitations, authors describe what limitations they realized in future research if CPP-1 as real compounds. There are several CPP-1 compounds reported by different groups (e.g., Feng and Zhang, 2020; Tang S et al 2021) in previous. I am not sure if their structural identity or slightly different depending on the extraction method. Wondering if they tested the analysis of CPP-1 as a validation study. For example, a proteomic study utilized size exclusion chromatography (SEC) coupled with light scattering (LS) detectors to determine the molecular size distribution or enzymatic digestion
In the Material method section,
There are missing statistical analyses, so, please describe the statistical method for determining the criteria following their power of statistical analysis.
Author Response
Our responses to Reviewer 5 comments:
Su Q. et al investigated the anti-obesity from natural resources like the root of Codonopsis pilosula which reflects well multi-physiological functions such as anti-inflammatory activity, anti-tumor activity, and anti-diabetic activity, hypoglycemic, Anti-viral, antioxidant, and prebiotic in animal models. In this report, they challenge to extract bioactive compounds and isolate them and characterize them by utilizing physical-chemical methodologies. To determine anti-obesity activity, this novel compound was examined in its biological functions using HFD-induced mice obesity followed by treatment of named CPP-1 novel compound under their report entitled “The obesity amelioration effect in high-fat-diet fed mice of a homogeneous polysaccharide from Codonopsis pilosula.” It is very interesting and neatly documented. However, several points need to be confirmed and validated by promoting going through Q/A sessions as follows.
- The authors mentioned they used n=10 mice without sex description in detail. So, I Was wondering about gender differences, and would be helpful if they describe it.
Response: Thanks for the comment of reviewers. All mice used in this research are male C57BL/6. We have added sex description in Materials and Methods-3.8. HFD-induced obesity is influenced by complex factors, including strains, sex and age of the animal, and it is widely accepted that sex may impact on the mouse obesity development that male mouse tends to gain more body weight and fat weight than female. In order to ensure that a clear obesity phenotype was obtained, we consulted a series of literatures and selected C57BL/6 male mice as in vivo model. This issue has been described in the revised manuscript on Line 259 as follows.
C57BL/6 male mice (eight-week-old) used in this experiment were obtained from HuaFukang BioScience Company (Beijing, China) and housed in SPF-grade according to requirements of the Institutional Ethics Committee of Shanghai Institute of Materia Medica (2021-04-XC-38).
- It would be great if they can provide a complete structure of CPP-1
Response: This comment is very helpful for us. Due to the impact of the COVID-19 pandemic in China, it was a problem to reserve the instrument. We will provide the complete structure of CPP-1 afterwards.
- Regarding hyperglycemic measurement, are there any parameters available to demonstrate the CPP-1 ameliorate fat tissue or systemic level? Is any histological data available to show CPP-1 as an anti-obesogenic effect in the C57Bl/6 mice model? For example, is there any difference in H&E staining using epididymal adipose tissues with serum biochemical analysis (e.g., TG, TC, plasma glucose, plasma insulin, HDL/LDL, etc)?
Response: In the revised manuscript, we provided more proofs about anti-obesity effect of CPP-1. On systemic level, CPP-1 treatment reduced the HFD-induced elevation of serum total cholesterol, ALT, AST. Meanwhile, CPP-1 treatment can also down-regulate hepatic triglyceride. Furthermore, CPP-1 supplementation improved morphology of liver and adipose tissues. The figures have been added in Fig.4B, E, F, G, I.
- What do they believe is the molecular target of CPP-1if it is supposed to be true CPP-1 is the candidate of anti-obesogenic compounds? Is there any evidence?
Response: CPP-1 is a macromolecular compound. We don't know its molecular target, but we have observed its weight loss effect on mice.
- Have authors measured any PKJ/PD value of CPP-1 in blood levels or have been measured half -of a lifetime (T2/1) of CPP-1 in blood?
Response: We did not measure the PKJ/PD value and T1/2 of CPP-1 in blood, because we think it is difficult for macromolecule CPP-1 to be directly absorbed into the blood.
- Is this CPP-1 circulating to the brain-blood barrier or not?
Response: We have not tested whether CPP-1 can enter the blood-brain barrier, but from the perspective of the large molecular weight and strong water solubility of CPP-1, it should be difficult to penetrate the blood-brain barrier.
- Table 1 methylation data, have they measured retention time depend on their detection condition which may need to be described in detail in the 3.6 at M & M.
Response: Thanks a lot for your kind comments. We have written the retention time in Table 1.
|
Retention Time |
Methylated |
Type of linkage |
Major Mass Fragments (m/z) |
Percentage (%) |
|
8.739 |
2,3,4,6-Me4-Glcp |
T-Glcp |
43,59,102,118,129,161,205 |
20.53 |
|
9.856 |
2,4,6-Me3-Glcp |
1,3-Glcp |
43,59,87,101,118,129,161,202,234,277 |
4.80 |
|
9.990 |
2,3,6-Me3-Galp |
1,4-Galp |
43,59,71,87,102,118,129,162,233 |
34.84 |
|
10.113 |
6-Me-Glcp |
1,2,3,4-Glcp |
87,99,115,129,157,185,218,160,333 |
10.36 |
|
10.698 |
2,3,4,6-Me4-Manp |
T-Manp |
43,59,102,118,129,161,205 |
5.17 |
|
10.933 |
2,6-Me2-Glcp |
1,3,4-Glcp |
43,59,87,118,129,160,185,305 |
15.25 |
|
12.308 |
2,3,5-Me3-Araf |
T-Araf |
59,71,87,102,118,129,161,162 |
9.05 |
Table 1. Linkage patterns analysis of CPP-1.
- Have they confirmed it by TLC analysis to confirm the sugar moiety of CPP-1
Response: Thanks for this point, TLC is a trace and rapid analytical method for the analysis of the monosaccharide composition of hydrolyzed polysaccharides. However, there are several methods to detect the structure of polysaccharides such as gas chromatography(GC), high performance liquid chromatography(HPLC) and ion chromatography(IC) to determine the monosaccharide composition. The monosaccharide composition of CPP-1 was determined by HPLC-DAD. HPLC has higher resolution and can perform both qualitative and quantitative analysis, and is now the main method for monosaccharide composition analysis; We chose to detect the monosaccharide component of CPP-1 by HPLC.
- In the title, the authors use homogeneous polysaccharides… is there any criteria for homogeneity in the final structure of CPP-1?
Response: Thanks a lot for your kind comments. A homogeneous polysaccharide refers to a polysaccharide sample with a continuous, normally distributed molecular mass and a defined monosaccharide composition and monosaccharide residue linkage. As shown in Figure 1B, CPP-1 shows a symmetrical peak, indicating that it is a homogeneous polysaccharide. The results indicated that CPP-1 was composed of mannose (Man), glucose (Glc), galactose (Gal) and arabinose (Ara) at a molar ratio of 5.86: 51.69 : 34.34: 8.08. The methylation analysis revealed that the main glycosidic linkage types of CPP-1 were (1→4)-linked-Gal residue, (1→3)-linked-Glc residue, (1→3,4)-linked-Glc residue, (1→2,3,4)-linked-Glc residue, (1→)-linked-Glc residue, (1→)-linked-Man residue and (1→)-linked-Ara residue.
- Limitations, authors describe what limitations they realized in future research if CPP-1 as real compounds. There are several CPP-1 compounds reported by different groups (e.g., Feng and Zhang, 2020; Tang S et al 2021) in previous. I am not sure if their structural identity or slightly different depending on the extraction method. Wondering if they tested the analysis of CPP-1 as a validation study. For example, a proteomic study utilized size exclusion chromatography (SEC) coupled with light scattering (LS) detectors to determine the molecular size distribution or enzymatic digestion In the Material method section.
Response: Thank you for the above suggestions. Regarding the question that the structural characteristics of CPP-1 compounds or slightly differ depending on the extraction method, the polysaccharide structures obtained by different extraction methods and isolation and purification of polysaccharides are different. The polysaccharides obtained by the same preparation method are identical. We ensured the consistency of the preparation method of CPP-1, which was determined to be all CPP-1 by structural characterization. The homogeneity and molecular weight of CPP-1 were determined by HPGPC combined with an ELSD detector. Our subsequent study attempted to determine the molecular size distribution in polysaccharides using size exclusion chromatography (SEC) coupled with a light scattering (LS) detector.
- There are missing statistical analyses, so, please describe the statistical method for determining the criteria following their power of statistical analysis.
Response: All data are shown as mean ± SEM. Statistical significance was determined using two-tailed Student’s t-test when comparing two groups. The p value less than 0.05 was considered statistically significant. We have added this part to the article. The Statistical Analysis have been added in Materials and Methods-3.14. Statistical Analysis (line 310-313).

Round 2
Reviewer 3 Report
-
Reviewer 4 Report
The authors have made a brilliant work in revisiting all part of the manuscript which has been substantially improved.
Line 482-485: Please consider spliting the sentence in two.
Kindly make a quick control on the references' format during proofs.